# Predicting biomass of rice with intermediate traits: Modeling method combining crop growth models and genomic prediction models

Yusuke Toda[1], Hitomi Wakatsuki[2], Toru Aoike[1], Hiromi Kajiya-Kanegae[3], Masanori Yamasaki[4], Takuma Yoshioka[4], Kaworu Ebana[5], Takeshi Hayashi[6], Hiroshi Nakagawa[3], Toshihiro Hasegawa[7], Hiroyoshi Iwata[1] *

1 Department of Agricultural and Environmental Biology, Graduate School of Agricultural and Life Science, The University of Tokyo, Tokyo, Japan, 2 Institute for Agro-Environmental Sciences, National Agriculture and Food Research Organization (NARO), Ibaraki, Japan, 3 Research Center for Agricultural Information Technology, NARO, Ibaraki, Japan, 4 Food Resources Education and Research Center, Graduate School of Agricultural Science, Kobe University, Hyogo, Japan, 5 Genetic Resources Center, NARO,, Ibaraki, Japan, 6 Institute of Crop Science, NARO, Ibaraki, Japan, 7 Tohoku Agricultural Research Center, NARO, Iwate, Japan

* aiwata@mail.ecc.u-tokyo.ac.jp

**Data Availability Statement:** All relevant data are available from the GitHub repository (https://

## Abstract

Genomic prediction (GP) is expected to become a powerful technology for accelerating the genetic improvement of complex crop traits. Several GP models have been proposed to enhance their applications in plant breeding, including environmental effects and genotype-by-environment interactions (G×E). In this study, we proposed a two-step model for plant biomass prediction wherein environmental information and growth-related traits were considered. First, the growth-related traits were predicted by GP. Second, the biomass was predicted from the GP-predicted values and environmental data using machine learning or crop growth modeling. We applied the model to a 2-year-old field trial dataset of recombinant inbred lines of japonica rice and evaluated the prediction accuracy with training and testing data by cross-validation performed over two years. Therefore, the proposed model achieved an equivalent or a higher correlation between the observed and predicted values (0.53 and 0.65 for each year, respectively) than the model in which biomass was directly predicted by GP (0.40 and 0.65 for each year, respectively). This result indicated that including growth-related traits enhanced accuracy of biomass prediction. Our findings are expected to contribute to the spread of the use of GP in crop breeding by enabling more precise prediction of environmental effects on crop traits.

## Introduction

Genomic selection (GS) [1] is a novel method increasingly being used in plant and animal breeding. Meuwissen et al. proposed the use of genomic prediction (GP) to predict genotypic

github.com/YT100100/ReferenceData_2018_
PLoSONE). Powered by

**Funding:** Author HI received funding from the
Japan Society for the Promotion of Science
(https://www.jsps.go.jp/index.html), KAKENHI,
grants JP25252002 and JP16H0458. Author HI
received funding from the Japan Science and
Technology Agency (https://www.jst.go.jp),
CREST, grant JPMJCR16O2. The funders had no
role in study design, data collection and analysis,
decision to publish, or preparation of the
manuscript.

**Competing interests:** The authors have declared
that no competing interests exist.

values (or breeding values) of selection candidates from whole-genome marker genotypes and
a statistical model [1]. GP enables the prediction of genotypic values of a target trait without
information about its causal genes, even when the target trait is controlled by a number of
genes with complex interactions. Recent falls in the cost of genotyping high-density genome-
wide markers have inspired the increased use of GP in animal breeding [2] and plant breeding
[3–5]. Because phenotypic values predicted by GP can be used as alternatives to phenotypic
values observed in field trials, GP can accelerate breeding by skipping field experiments for
selection, and thus is expected to increase selection gains per unit time [6].

Because environmental effects, i.e., the main effects of the environment and of the geno-
type-by-environment interaction (G×E), are generally not trivial in plant breeding, the use of
GP models without consideration of these effects can cause difficulties in the application of GP
to yield-related traits, which can be strongly influenced by these effects [7]. Several methods
have been proposed to consider environmental effects, including the modeling of covariance
between genotype and environment [8–9], consideration of marker-by-environment interac-
tions [10], and inclusion of environmental covariates [11]. Moreover, a GP model that can
take environmental effects into account will benefit the application of GS in plant breeding
because it will lead to more accurate predictions of genetic values for yield-related traits under
a target environment and thus to a higher genetic gain per cycle [6].

Crop growth models (CGMs) are expected to be an important tool for plant breeding
because they incorporate environmental effects into the GP framework [12–13]. For example,
Heslot et al. [14] used a CGM to select the environmental covariates which were included in a
GP model. Technow et al. [15] proposed a method for integrating a CGM and a GP model
with approximate Bayesian computation, and Cooper et al. [16] applied the method to maize
data. However, the models in these studies attained only a small improvement in accuracy
when applied to real data. One of the reasons for the small improvement may be the difficulty
in parameter estimation of CGMs. The accurate estimation of CGM parameters is difficult
when it is only based on observations of a target trait. In other words, the accuracy can be
improved when observation of traits related to the target traits is included in the parameter
estimation of CGM.

The growth-related traits may be good candidate traits to improve the prediction accuracy
of target traits. Several studies have used growth-related traits with multi-trait GP models to
improve the prediction accuracy of target traits [17–18], suggesting that the growth-related
traits convey precise growth details and provide useful information for target trait prediction.
To date, there has been no research that used growth-related traits for CGM and GP integra-
tion. In this study, we proposed a method to use the phenotypic data of growth-related traits
in the integrated models of GP and CGM. This method has two steps. First, the growth-related
traits are treated as "intermediate traits" and are predicted by GP. Second, the target traits are
predicted from the predicted values of the "intermediate traits" and environmental data using
a CGM. By dividing the model into two steps that correspond to GP and CGM, the "interme-
diate traits" can be naturally included into the model without complex statistical modeling of
the relation between GP and CGM.

To validate this integrated model, rice is a suitable research species because there have been
previous studies of the application of GP [19–24] and CGMs [25–27], such as SIMRIW [28]
and CERES-rice [29]. However, attempts to integrate these methods to predict phenotypic var-
iations in rice have been lacking, with some exceptions [30]. Biomass is also a suitable trait for
validation. Biomass is a direct target of breeding for biofuel rice [31–32] and is an important
component of grain yield [33–34].

In this study, we developed models to predict the biomass of rice, in which the observed
phenotypic data of growth-related traits, whole-genome marker genotype, and environmental

data were used. The model comprised two steps wherein the intermediate traits were predicted with GP in the first step and biomass was predicted from the predicted values of the intermediate traits in the second step. In the intermediate traits, the heading date is exceptionally predicted using a development rate (DVR) model based on the data obtained from multi-environmental trials (METs) and the genotypes of heading-date-related markers. Additionally, in the second step, we evaluated the potential of a "black box"-type machine-learning model, in which a detailed model structure was not defined as a priority for substituting the CGM.

These models were validated with a recombinant inbred line (RIL) population of japonica rice for biomass prediction. We conducted 2-year field experiments of the population. The experiments were conducted with different timings of sowing (and planting) between both years to evaluate the potential of the models under different environments. The difference in sowing (and planting) dates was about one month, and this caused different phenological developments of the plants between those two years. Finally, the models were evaluated for their accuracy of biomass prediction within the experiments (using the same-year experiment for training and validation) and between the experiments (using one year's experiment for training and the other year's experiment for validation).

## Materials and methods

### Plant materials

We evaluated 123 RILs derived from a cross between two *japonica* cultivars—Koshihikari and Kinmaze—and both parental lines. The construction of RIL was in the $F_8$ generation in 2014 and in the $F_9$ generation in 2015. Because Kinmaze and Koshihikari have different growth patterns and plant structure, these RILs were expected to be suitable for analyzing genetic variations observed in growth differences. In 2014 and 2015, experiments were conducted in an experimental paddy field of the National Agriculture and Food Research Organization, Tsukuba, Ibaraki, Japan (36° 01' N, 140° 06' E, 22m above sea level). Sowing and transplanting were performed in different months between years to produce results under different conditions of day length and temperature; we sowed seeds on 22 April 2014 and 19 May 2015 and transplanted seedlings into the field on 20 May 2014 and 18 June 2015. Because of different cultivation periods during 2014 and 2015, the 2-year experiments were not simply yearly replications but were expected to induce different growth patterns under different environmental conditions. Plants were transplanted 15 cm apart in rows 30 cm apart in plots. We transplanted two seedlings per hill. The area for each line per replicate was 60 cm × 105 cm (2 rows × 7 hills). Inorganic fertilizer (80–100–100 kg of N-P2O5-K2O ha−1) was applied to the field. Aboveground plant organs were harvested to determine biomass at physiological maturity, which spanned from 29 August to 10 October in 2014 and from 17 September to 5 November in 2015 depending on variation among lines. Dry matter weight above ground was used as biomass.

We recorded leaf age and number of tillers on each of several dates to evaluate variations in the growth pattern of the RILs (Table 1). The leaf age is calculated using the following formula [35]:

$$\text{Leaf age} = \text{Number of developed leaves} + \frac{\text{Length of the developing leaf}}{\text{Final length of the developing leaf}}$$

We used leaf age instead of leaf number to treat the development of leaves as continuous values. The maximum tiller number was determined on the basis of measurements of the tiller number observed at three and five different time points in 2014 and 2015, respectively. The measurements were continued until the leaf number on the main culm reached to 11 or more.

**Table 1. Dates of observation of leaf age and number of tillers.**

| Year | Sowing date | Year | Dates |
|---|---|---|---|
| 2014 | 22-Apr | Leaf age | 5/19, 6/2, 6/9, 6/16, 6/23, 6/30, 7/7, 7/14, 7/22, 7/28, 8/4 |
| | | Number of tillers | 6/9, 6/16, 6/23 |
| 2015 | 19-May | Leaf age | 6/15, 6/25, 7/2, 7/9, 7/16, 7/23, 7/30, 8/5, 8/10, 8/17, 8/24, 8/31 |
| | | Number of tillers | 6/25, 7/2, 7/9, 7/16, 7/23 |
| Trait | Year | Dates | |
| Leaf age | 2014 | 5/19, 6/2, 6/9, 6/16, 6/23, 6/30, 7/7, 7/14, 7/22, 7/28, 8/4 | |
| | 2015 | 6/15, 6/25, 7/2, 7/9, 7/16, 7/23, 7/30, 8/5, 8/10, 8/17, 8/24, 8/31 | |
| Number of tillers | 2014 | 6/9, 6/16, 6/23 | |
| | 2015 | 6/25, 7/2, 7/9, 7/16, 7/23 | |

This was because our preliminary experiments with nine diverse cultivars suggested that the tiller number reached its maximum before 11 leaves were observed.

Length of the fully expanded leaf blades was measured for the 5th leaf, 11th leaf, flag leaf and 2 leaves below the flag leaf. According to our preliminary study, the final length of the leaf blade on the main culm increased almost linearly with the leaf age from 5 to 11. The increment in the final length per leaf age ($\Delta$LL) was derived from the 5th and 11th leaves. Leaf age, number of tillers and leaf blade length on the main culm were recorded for two plants per entry for each replicate. Heading date and biomass were recorded on 6 plants per entry.

We used a method similar to [36] for the genotyping of RILs by extracting DNA from bulked seedlings of each $F_7$ line (corresponding to the $F_6$ generation) via a CTAB-based extraction method [37]. We used single-nucleotide polymorphism (SNP) markers for the linkage map construction, and a total of 703 SNPs were selected from genome-wide SNP data [38–39] and analyzed using a BeadStation 500G system (Illumina, CA, USA) according to the manufacturer's instructions. Finally, using R software [40] and the R/qtl package [41], we deleted SNPs with identical genotypes with the findDupMarkers function. Finally, a total of 315 SNPs were used for the genotyping of RILs (S1 Fig).

Air temperature and solar radiation were recorded on-site (available at http://www.naro.affrc.go.jp/org/niaes/aws/). Photosynthetically active radiation (PAR) was estimated from the solar radiation assuming that proportion of PAR to the global solar radiation is 0.5 [42]. Daily means of temperatures are shown in Fig 1.

Because the RIL population was cultivated in only one field, it was difficult to estimate model parameters for heading date in CGM. To obtain the model parameters, we used heading dates recorded in METs that tested 112 cultivars, including Kinmaze and Koshihikari, most of which were developed in Japan. METs were conducted in six locations in several years (33 trials, Table 2).

## Genetic analysis of observed traits

All statistical analyses were conducted in R software [40]. The arithmetic means of observed values were used as phenotypic values for each RIL in the following analysis. The number of replications for each trait was described in the previous section. Analysis of variance (ANOVA) was conducted to evaluate the significance of genotype and environmental effects and their interaction.

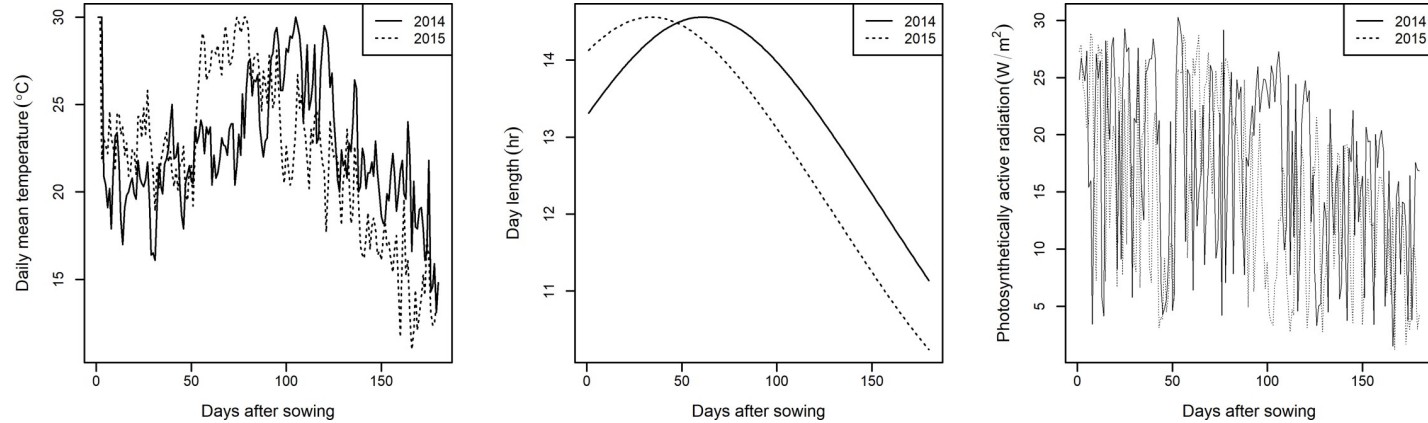

**Fig 1. Environmental data during growing season.** Daily mean temperature, theoretical day length and photosynthetically active radiation (PAR) of Tsukuba under field trial of RILs are shown. Data in both 2014 and 2015 are expressed as solid and dotted lines, respectively.

We evaluated the accuracy of GP of all traits with 10-fold cross-validation. For building the GP models, we employed four methods. Two of them were regularized regression: ridge regression (RR) and LASSO, and the other two were Gaussian process regression (reviewed by [43]): one based on an additive relationship matrix (GBLUP) and the other on a Gaussian kernel matrix (RKHS) as a representative of covariance matrix. We used the "glmnet" package [44] for RR and LASSO, and the "rrBLUP" package [45] for GBLUP and RKHS. The narrow-sense heritability of each trait was estimated using a mixed model based on an additive relationship matrix in GBLUP.

## Growth process analysis

We analyzed the change in leaf age and the number of tillers during growth as a simple function of heat units (accumulated daily mean temperature). The leaf age and the number of tillers on the $i$th day from sowing ($\mathrm{Leaf}_i$, $\mathrm{Till}_i$, dimensionless) were represented as:

$$\mathrm{Leaf}_i = \min(\Delta\mathrm{Leaf} \times \mathrm{HU}_i, \mathrm{Leaf}_{\mathrm{MAX}}) \tag{1}$$

$$\mathrm{Till}_i = \begin{cases} 1 & (\mathrm{HU}_i \le 800) \\ \min(\Delta\mathrm{Till} \times \mathrm{HU}_i, \mathrm{Till}_{\mathrm{MAX}}) & (\mathrm{HU}_i > 800) \end{cases} \tag{2}$$

where $\mathrm{HU}_i$ (˚C) represents heat unit ($\Sigma$ daily mean temperature from emergence to the $i^{\mathrm{th}}$ date); $\Delta\mathrm{Leaf}$ (˚$\mathrm{C}^{-1}$) and $\Delta\mathrm{Till}$ (˚$\mathrm{C}^{-1}$) represent the rate of change per HU; and $\mathrm{Leaf}_{\mathrm{MAX}}$ and $\mathrm{Till}_{\mathrm{MAX}}$ represent maximum values. Because we observed the growth of each line, $\Delta\mathrm{Leaf}$ and $\Delta\mathrm{Till}$ were estimated as slopes of linear regression of phenotypic data during the study period,

**Table 2. Location, year, and number of replications of field experiments to record heading date.**

| Location | 2004 | 2005 | 2006 | 2007 | 2008 | 2009 | 2010 | 2011 | 2012 | 2013 | 2014 |
|---|---|---|---|---|---|---|---|---|---|---|---|
| Daisen, Akita | | | | | | | | | 1 | 1 | 1 |
| Tsukuba, Ibaraki | | | | | 1 | 1 | 2 | 3 | 3 | 1 | 1 |
| Tsukubamirai, Ibaraki | 2 | 2 | | | | | | | | | |
| Kasai, Hyogo | | | 1 | 1 | 1 | 1 | 1 | 1 | 1 | 1 | 1 |
| Fukuyama, Hiroshima | | | 1 | 1 | 1 | 1 | 2 | 2 | 2 | 2 | 2 |
| Fukuoka, Fukuoka | | | | | | | | | 1 | 1 | 1 |

whereas Leaf$_{MAX}$ and Till$_{MAX}$ were measured at the end of the growth period. Because leaf age and number of tillers are generally not considered linear to HU, we assumed that its growth was approximated by a combination of linear functions.

Generally, the growth of rice does not proceed when the daily temperature is low. To take this assumption into consideration, we developed the growth models of leaf age and number of tillers based on the heat unit, in which the base temperature of the growth of rice was considered ($\Sigma$max(0, daily mean temperature–8˚C)) instead of the simple heat unit. The lower bound of temperature was obtained from [42]. However, the result did not largely differ or was even more inaccurate in the prediction accuracy than models developed based on the simple heat unit. Thus, we present only the results based on the simple heat unit in this paper.

## Prediction of heading date by DVR model

To predict heading date in a target environment, we used Yin et al.'s model [46] modified by Nakagawa et al. [47], which describes daily developmental rate (DVR) as a function of environmental factors (DVR model, hereafter). In the DVR model, daily progress of a developmental stage is expressed as a continuous value representing developmental stage (DVS), ranging from 0 (emergence) to 1 (heading). The DVS at the $n$th day after emergence is the sum of the daily development rates (DVR$_i$):

$$\text{DVS}_n = \sum_{i=1}^{n} \text{DVR}_i \tag{3}$$

where DVR$_i$ is given by daily mean temperature ($T_i$, ˚C) and day length ($P_i$, h):

$$\text{DVR}_i = \begin{cases} \dfrac{f(T_i)^\alpha g(P_i)^\beta}{G} & (\text{if } 0.145 + 0.005G \leq \text{DVS} \leq 0.345 + 0.005G) \\ \dfrac{f(T_i)^\alpha}{G} & (\text{if DVS} < 0.145 + 0.005G, 0.345 + 0.005G < DVS) \end{cases} \tag{4}$$

$$f(T_i) = \begin{cases} \dfrac{T_i - T_b}{T_o - T_b} \left( \dfrac{T_c - T_i}{T_c - T_o} \right)^{\frac{T_c - T_o}{T_o - T_b}} & (if \ T_b \leq T_i \leq T_c) \\ 0 & (if \ T_i < T_b, T_c < T_i) \end{cases} \tag{5}$$

$$g(P_i) = \begin{cases} \dfrac{P_i - P_b}{P_o - P_b} \left( \dfrac{P_c - P_i}{P_c - P_o} \right)^{\frac{P_c - P_o}{P_o - P_b}} & (if \ P_i \geq P_o) \\ 1 & (if \ P_i < P_o) \end{cases} \tag{6}$$

Six parameters were fixed ($T_b = 8˚C$, $T_o = 30˚C$, $T_c = 42˚C$, $P_b = 0h$, $P_o = 10h$, $P_c = 24h$) among lines as in [46]. The parameters $\alpha, \beta, G$ represent sensitivity to temperature, sensitivity to day length, and growth period, respectively, and are assumed to have specific values for each line. We estimated the values from the MET data using particle swarm optimization [48], which is used to optimize non-linear functions (Experiment A in Fig 2).

To calculate the values of $\alpha, \beta, G$ of the target RILs, we constructed models to predict them from marker genotypes (Experiment A in Fig 2) of six heading-date–related genes (*Hd1*, *Hd3a*, *Hd6*, *Hd16*, *Hd17*, and *Ghd7*) [49–54] of 112 lines. We used Extreme Learning Machine (ELM) [55], which is a machine learning method based on a neural network with advantages in generalization performance and learning speed, to model the relationships between the parameter values and the marker genotypes. After modeling these relationships, we estimated

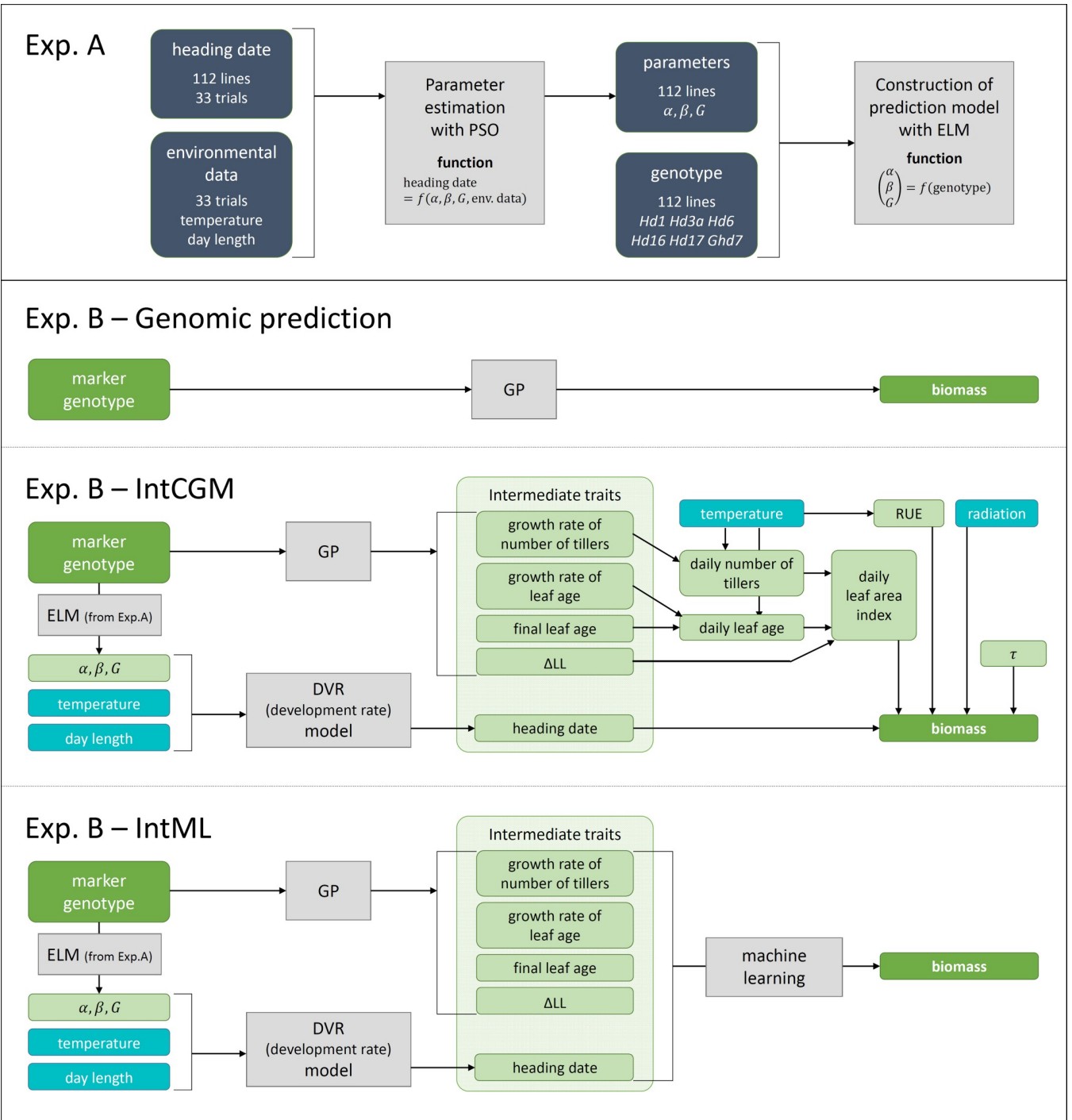

**Fig 2. Flow chart of model structures.** Experiment A: Process for estimating values of three parameters (α, β, G) related to heading date. Multi-environment trial data of heading date of 112 lines were used to model the relationship between parameter values and marker genotypes of heading-date-related genes using Extreme Learning Machine (ELM). Experiment B: Structure of conventional genomic prediction (GP), integrated CGM (IntCGM), and integrated machine-learning (IntML) models.

the values of α,β,G of the RILs by using the ELM prediction model (Experiment B in Fig 2). The marker genotypes of the heading-date–related genes of the RILs were assumed to be the same as those of the SNP nearest to the genes, and were used as inputs of the ELM model.

## Integrated GP–CGM model

We included environmental effects in the model of yield-related traits by integrating the GP models and a CGM proposed by [42], with modifications, to create an integrated CGM (IntCGM).

IntCGM has two steps (Experiment A in Fig 2). First, the GP and DVR models predict "intermediate traits" related to biomass. LASSO was selected as a representative GP model because it showed the highest accuracy among all the GP models in 10 of 14 traits (i.e., six intermediate traits and biomass in two years). Second, the CGM simulates the daily change in biomass from the "intermediate traits".

Total biomass (BM, g m$^{-2}$) was estimated as the product of the total biomass at the day of termination of seed growth (BM$_{TSG}$, g m$^{-2}$) and a technical coefficient $\tau$ (dimensionless):

$$BM = \tau\, BM_{TSG} \tag{7}$$

where $\tau$ represents the influence of factors that are not included in the model (e.g., precipitation, nutrient condition, disease) [27]. The parameter $\tau$ was estimated as an average of the ratio of BM$_{TSG}$ and observed BM when the prediction was conducted. The day of termination of seed growth was presumed to be the day when the accumulation of daily mean temperature after heading date reached 630°C [42]. BM$_{TSG}$ was calculated as the sum of daily increases of biomass:

$$BM_{TSG} = \sum_{i=1}^{TSG} RUE_i \times FINT_i \times PAR_i \tag{8}$$

where TSG is the day of termination of seed growth, FINT$_i$ is fraction of PAR intercepted by canopy of $i^{th}$ day (dimensionless), RUE$_i$ is radiation use efficiency (g MJ$^{-1}$), PAR$_i$ is photosyntically active radiation (MJ m$^{-2}$). RUE$_i$ is the product of the maximum RUE (IRUE = 2.2 g MJ$^{-1}$) and the ratio of actual daily RUE to IRUE (TRFRUE$_i$, dimensionless):

$$RUE_i = IRUE \times TRFRUE_i \tag{9}$$

where TRFRUE$_i$ is a function of daily mean temperature ($T_i$) (Soltani and Sinclair, 2012):

$$TRFRUE_i = \begin{cases} \dfrac{T_i - 10}{15} & (10 < T_i \leq 25) \\ 1 & (25 < T_i \leq 32) \\ \dfrac{T_i - 42}{10} & (32 < T_i \leq 42) \\ 0 & (otherwise) \end{cases} \tag{10}$$

FINT$_i$ is estimated from the leaf area index, LAI$_i$ (dimensionless), and the extinction coefficient ($k$ = 0.6):

$$FINT_i = \exp(1 - k LAI_i). \tag{11}$$

Although IRUE and $k$ are known to have variation among lines and environments [42], they are assumed to be constant in this study because of the difficulty in the estimation of IRUE and $k$ for each line and environment. LAI$_i$ is expressed as:

$$LAI_i = \left\{ \beta\, Till_i \sum_{l=1}^{Leaf_i} (l \times \Delta LL)^2 \right\} / S \tag{12}$$

where $\Delta LL$ (m) is the increase of leaf length per unit increase of leaf age, $\beta$ = 0.003 is a technical coefficient explaining shape of leaves and $S$ = 225 cm$^2$ is the ground area of one plant. Thus,

$l \times \Delta\text{LL}$ represents the length of a leaf in one node, which came out in $l$th order, and $\sum_{l=1}^{\text{Leaf}_i} (l \times \Delta\text{LL})^2$ is expected to be proportional to leaf area of one tiller.

## GP model integrated with machine learning

We also constructed a model replacing the CGM with a machine learning method. This integrated machine-learning model (IntML) has the same two-step structure as IntCGM, but the second step uses machine learning methods. In the second step, we built machine learning models that use intermediate traits as explanatory variables to predict biomass. We chose a multiple regression model as a linear machine-learning method (IntML1) and the Random Forest [56] model as a non-linear method (IntML2). The R package "randomForest" [57] was used to build the Random Forest prediction models. When building the model, the parameter "mtry" was set as 2 and the other parameters were set as default.

## Model validation

To evaluate the ability of the models to predict biomass, we used 10-fold cross-validation among genotypes. We also predicted tested (i.e., training) and untested (i.e., validation) environments. In the prediction of the tested environment, the data from the same year were used as both training and validation data; that is, biomass of a fold in one year was predicted from the data of the remaining folds and environmental data in the same year. This assumption corresponds to the situation in which we want to predict the biomass of untested lines in tested environments. In the prediction of the untested environment, data from different years were chosen as training and validation data; that is, biomass of a fold in one year was predicted from the data of the remaining folds and environmental data in the other year. This assumption corresponds to the situation in which we want to predict the biomass of untested lines in untested environments.

We calculated three statistics to measure prediction accuracy. The correlation coefficient of observed versus predicted values ($r$) is a measure of strength of relative relation between both values. The root mean squared error (RMSE) expresses the discrepancy between predicted and observed values. The regression coefficient of observed versus predicted values (slope) is a measure of shrinkage in the predicted values over the observed values. Observed and predicted values were used as dependent and independent variables, respectively. When predicted values approach observed values, $r$ and slope approach 1 and RMSE decreases. We repeated cross validation of 100 replicates for each combination of models and prediction schemes to estimate the standard deviation of indices ($r$ and slope) of prediction accuracy. The Steel–Dwass test, a nonparametric multiple comparison test, was performed to examine significant differences in prediction accuracy.

## Results

### Growth patterns and correlations among traits

Growth curves and fitted models of leaf age and number of tillers are shown in Fig 3. The results indicated that the models could express the growth of each trait despite their simplicity.

The comparison of phenotypic values between the two years of experiment is shown in Fig 4. Among estimated parameters of the growth models, strong correlations between the years were observed in Leaf$_\text{MAX}$ and heading date whereas weak correlations were observed in Till$_\text{MAX}$ (Fig 4). However, the distributions of $\Delta$Leaf and $\Delta$Till differed between the years. The ranges of phenotypic values of the heading date (e.g., minimum values were ca. 90 and 80 days in 2014 and 2015, respectively) and biomass also differed between the years, despite their high

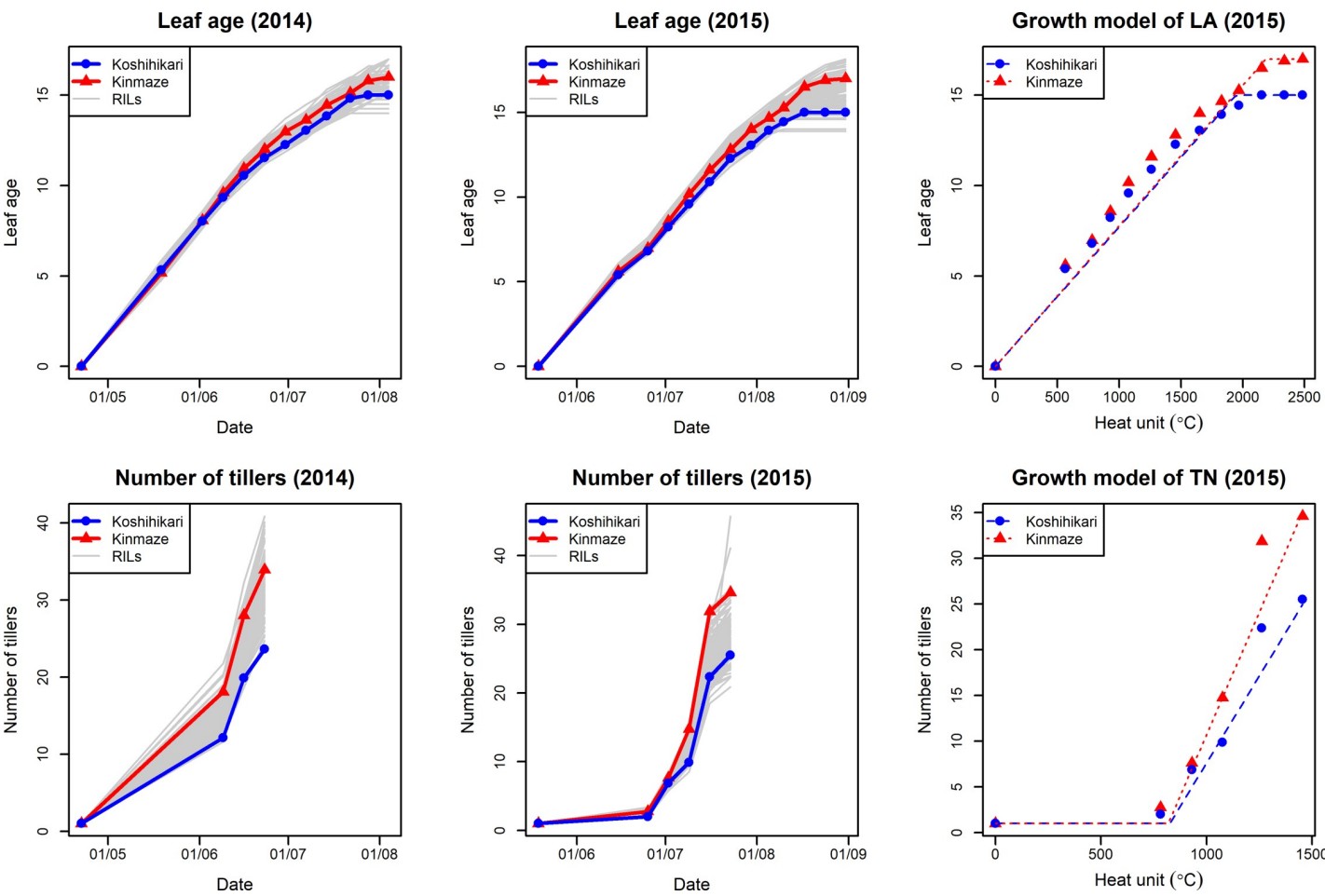

**Fig 3. Growth curves and growth models of leaf age and number of tillers.** The line means of both traits in 2014 and 2015 are plotted in four figures in the left side. The parents, Koshihikari and Kinmaze, and RILs are expressed as blue, red and gray lines, respectively. The growth models of those traits are shown in two figures in the right side. The growth model and the observed values of parents in 2015 are shown. Heat unit is used as horizontal axes.

correlations. The G×E effect was found to be significant ($p < 0.01$) for all traits using ANOVA. The correlation coefficients between growth-related traits and biomass were higher in 2015 than in 2014.

## Genomic prediction of growth-related traits

We assessed the prediction accuracy of the GP models (Fig 5) in growth-related traits, which corresponded to the first step of integrated models (IntCGM and IntML, Fig 2B). Accuracy was higher in 2015 than in 2014. Traits that showed higher correlation between years in Fig 4 tended to have higher values both in heritability and prediction accuracy. In ΔTill and Tillmax, the accuracy was lower than in biomass. In the following analyses, we chose LASSO as a representative GP model because it showed the highest accuracy among the models in 10 of 14 traits (six intermediate traits and biomass for two years). For heading date, we compared five models: the DVR model which used weather data and genome-wide marker data as explanatory variables and 4 GP models used only genome-wide marker data. The prediction accuracy was slightly lower in the DVR model than that in GP.

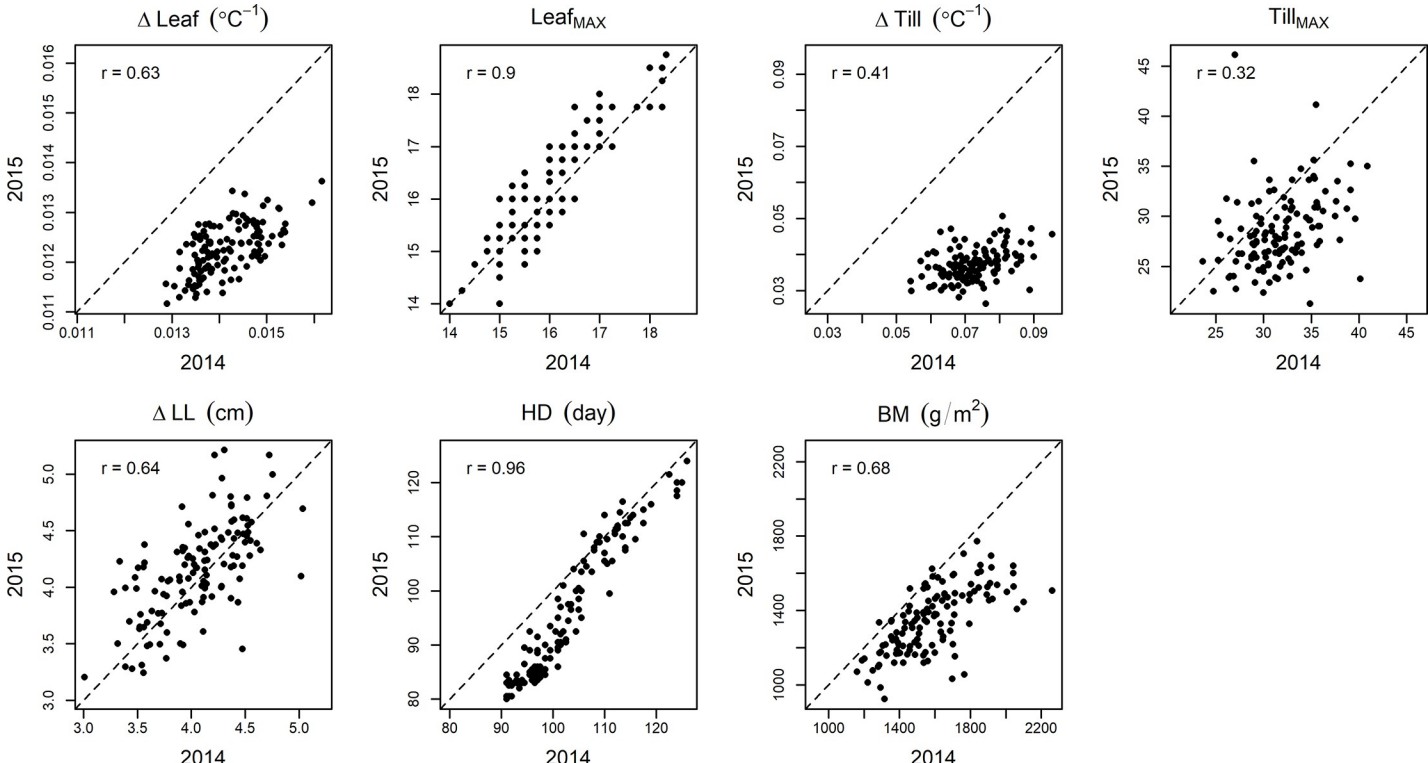

**Fig 4. Comparison of observed traits between 2014 and 2015.** Estimates of correlation coefficients between phenotypes of two years are shown in the top-left of each box. Abbreviations: ΔLeaf, growth rate of leaf age; Leaf$_{MAX}$, final leaf age; ΔTill, growth rate of number of tillers; Till$_{MAX}$, maximum number of tillers; ΔLL, growth rate of leaf length per leaf age; HD, heading date; HI, harvest index; BM, biomass.

## Prediction of biomass

In the tested environment, IntCGM, IntML, or both were more accurate at biomass prediction than GP with LASSO by all three statistics (Fig 6A), especially when the 2014 dataset was used as validation data: that is, IntCGM and IntML gave higher *r* values and regression slopes closer to one than GP, and IntML gave lower RMSE than GP. This tendency was supported by the fact that differences between *r* and slope of our models and those of GP were all statistically significant ($p < 0.01$).

IntCGM, IntML, or both performed better than or the same as GP in the untested environment (Fig 6B); both models gave significantly higher *r* and slope than GP except when IntML2 was tested with 2014 dataset as validation. IntCGM had a lower RMSE than that of GP using the 2015 dataset for validation but had a higher RMSE than that of GP using the 2014 dataset for validation.

We attempted to predict the panicle weight with IntCGM, wherein the panicle weight was expressed as the multiplication of biomass and harvest index and the harvest index was predicted using GP. However, the prediction accuracy of IntCGM was worse than GP because the harvest index itself was largely affected by the environment (S2 Fig).

## Discussion

### Accuracy of prediction of biomass

The *r* in our new models was the same as, or higher than, that of the conventional GP in the prediction of biomass (Fig 6). There was a substantial difference in the *r* of GP between 2014

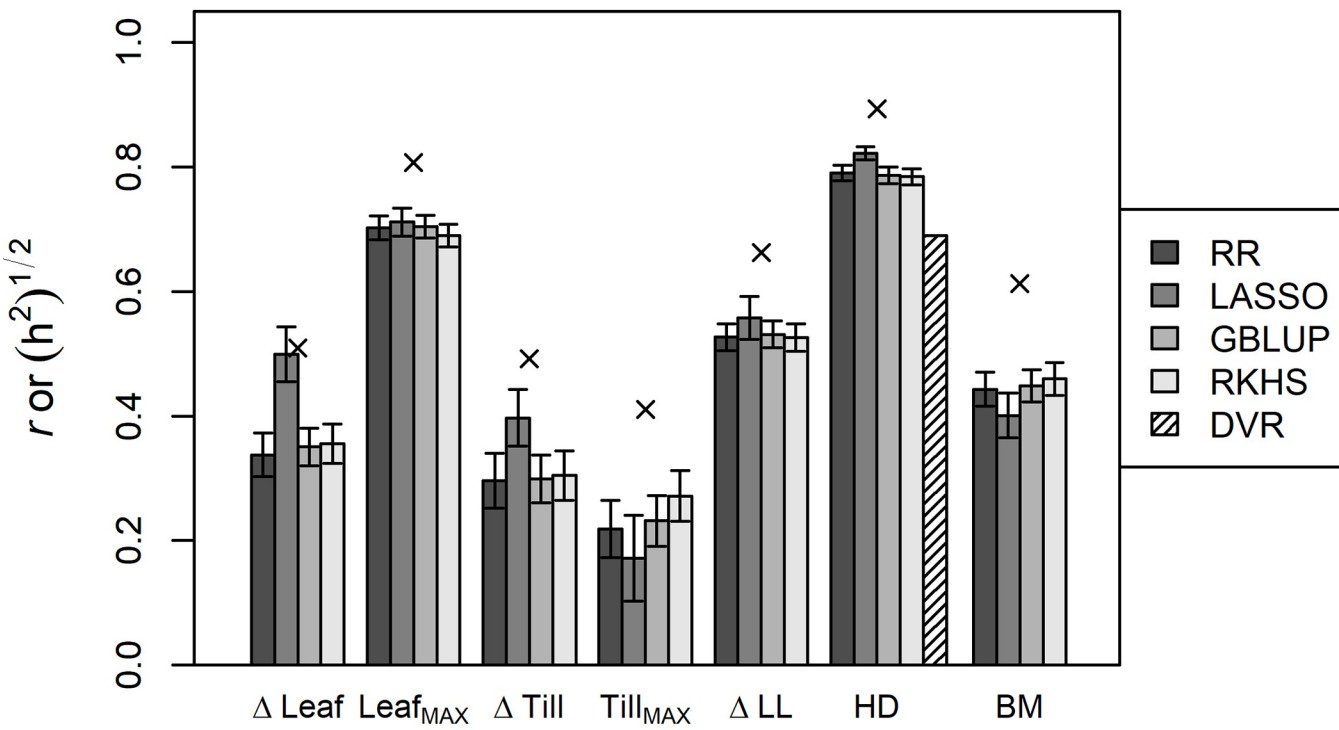

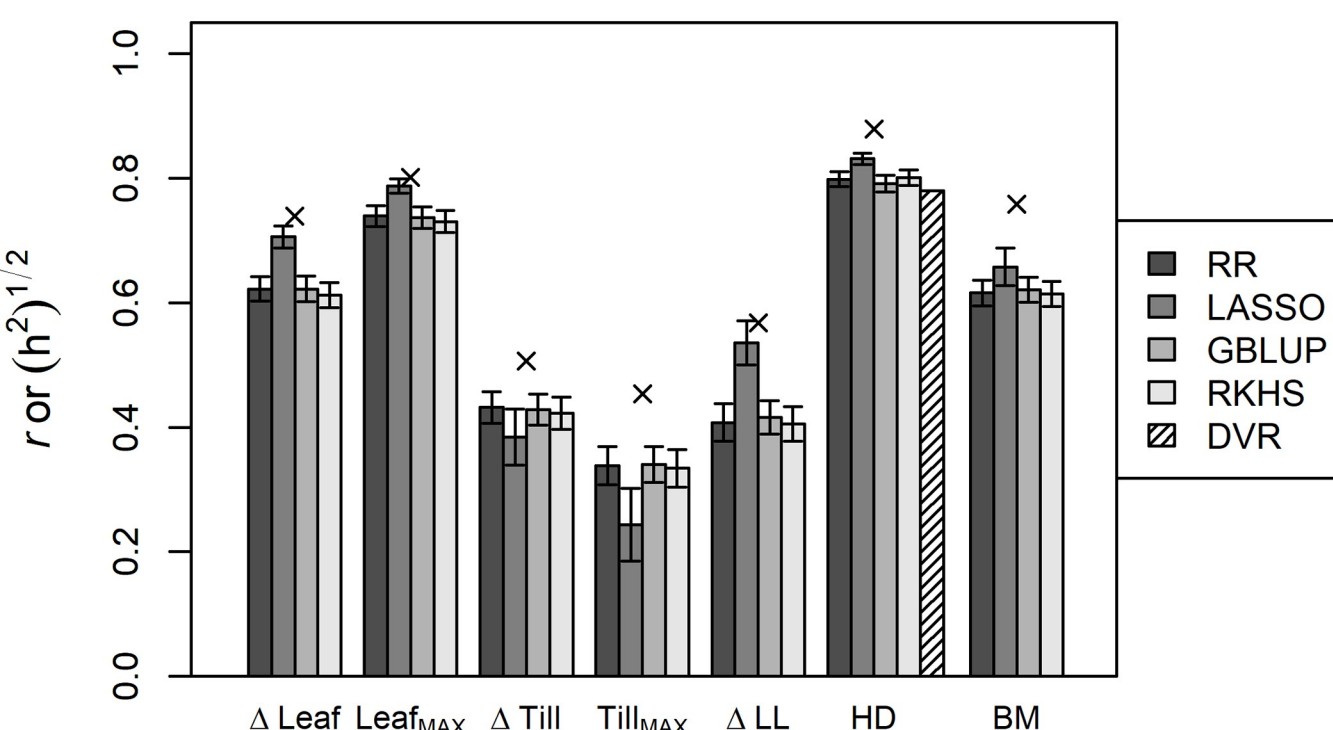

**Fig 5. Comparison of prediction accuracy of GP and heritability in growth-related traits.** Estimated correlation coefficients of observed values and values predicted using the five models for seven growth-related traits are shown as bars. The five models included four methods of whole-genome prediction (for all traits) and a DVR model with marker genotypes of the heading-date–related genes (for heading dates). The square roots of heritability of the seven traits are shown as crosses. Error bars represent ± 1 s.d.

and 2015 in the prediction of the tested environment, indicating that there was a difficulty in explaining the variation of biomass in 2014 through the direct linear regression of the genotypic markers. In contrast, the integrated models showed the significant increase in $r$ compared with that of GP in the 2014 prediction. These results indicate that the use of the intermediate traits was beneficial for improving accuracy of biomass prediction. Heading date prediction, which showed high heritability in both years, mostly contributed to the improved prediction accuracy.

Focusing on the GP trained with biomass of 2014, the accuracy was higher in biomass prediction of 2015 than in that of 2014. This intuitively unexpected result might be owing to two reasons. One is the low heritability of biomass in 2014, which led to lower prediction accuracy in the models [58–59]. To reduce the influence of the heritability level on the index of the prediction accuracy (i.e., a correlation coefficient between observed and predicted phenotypes), the value of $r$ was adjusted by dividing it by the square root of genomic heritability. The adjusted values of $r$ became 0.652 and 0.746 for the biomass in 2014 and 2015, respectively, and had smaller differences than the previous $r$. Another reason for the higher biomass prediction accuracy in 2015 is the GS model built with LASSO. In Fig 5, the biomass prediction accuracy was lower in LASSO than in other models in 2014, whereas the result was the opposite in 2015. Polygenic marker effects seemed more dominant in biomass in 2014 than in 2015 because LASSO is not good at capturing the small effects of a large number of variables. In contrast, the estimation of genomic heritability effectively reflects polygene effects. The differences in the characteristics of each estimation method subsequently caused the difference in the adjusted values of $r$ for the biomass in 2014 and 2015.

Although heading date was predicted by ELM and DVR models in our models, the prediction accuracy was worse than that by GP. One possible reason is that the heading date of RILs that we used could not be completely explained by heading-date-related genes, (i.e., *Hd1*, *Hd3a*, *Hd6*, *Hd16*, *Hd17*, and *Ghd7*) considered in ELM and DVR models. However, we employed the DVR model in our models because it can be used to predict the heading date in a new environment.

## Comparison with models in other studies

An advantage of our new approach over conventional researches of integrated models of GP and CGM is the inclusion of observed growth data in the model as "intermediate traits". This enables us to treat parameters in the model as representations of actual crop conditions. Two studies designed to integrate a genomic prediction model with a crop model [15, 19] tried to estimate growth parameters by using only phenotypic values of target traits. Technow et al. integrated GP and CGM to predict the yield of maize using parameter estimation with the approximate Bayesian computation [15]. Onogi et al. also constructed an integrated model to predict the heading date of rice [19]. However, this approach is difficult to apply to a complex trait, such as yield, and did not improve the prediction accuracy when it was applied to real-yield data [16]. It is also difficult to validate the accuracy of the estimated growth parameters. The use of the intermediate traits was beneficial for improving prediction accuracy and for further understanding how the parameters influence the target traits.

A multi-trait GP is another approach to predict target traits with intermediate traits (or secondary traits). In this model, the covariance structure among target and intermediate traits is

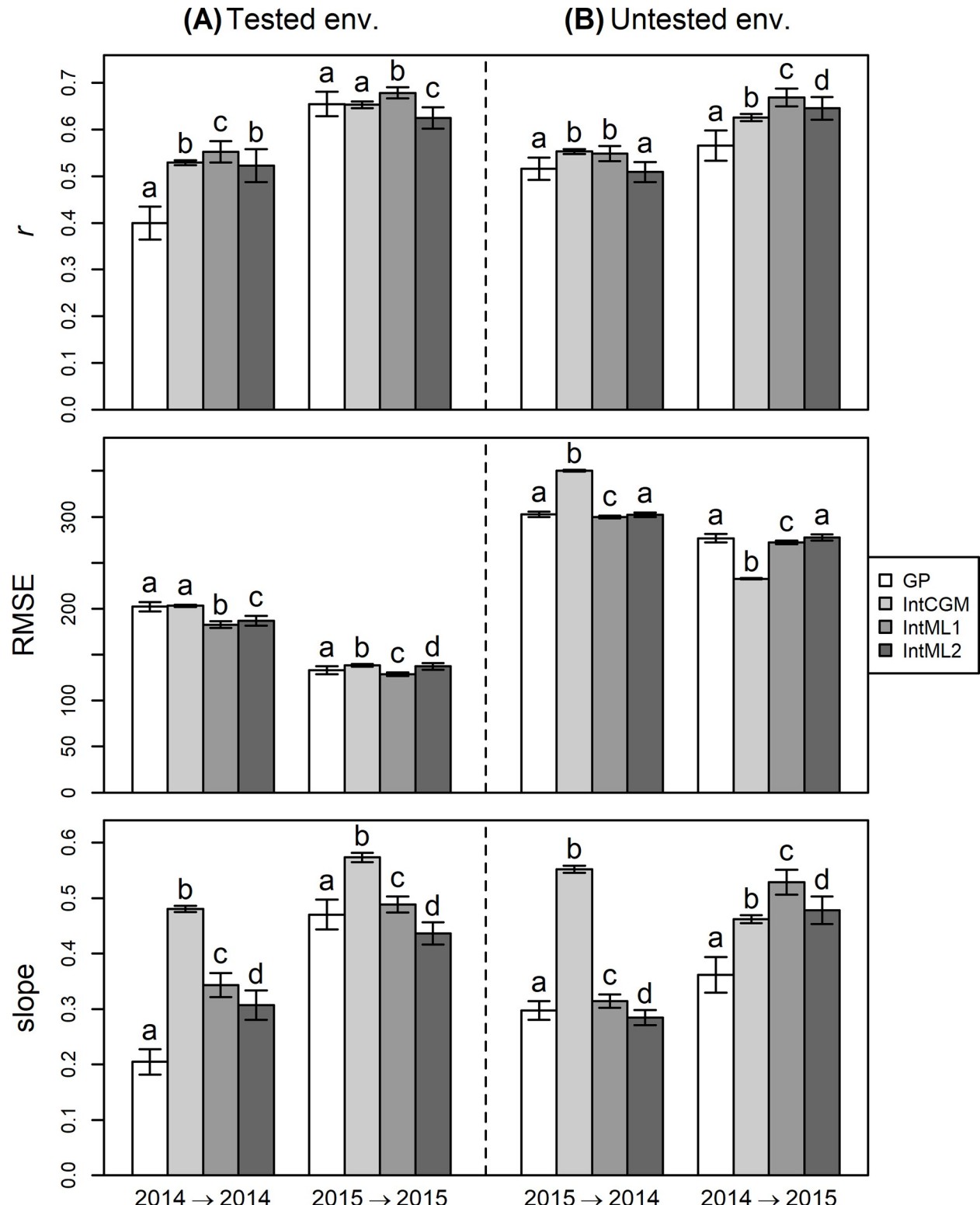

**Fig 6. Comparison of prediction accuracy of biomass.** Result of prediction of tested environment (A) and untested environment (B) are shown. LASSO was chosen as a representative GP model. Three indices are used: Correlation coefficient (r), RMSE (root mean squared error), and slope of the regression line for predicted and observed values. Error bars represent ± 1 s.d. Letters above the bars indicate a significant difference as determined by the Steel–Dwass test ($p < 0.01$).

considered to improve prediction accuracy [60–61]. For example, there are studies in which longitudinal traits measured by remote sensing were used as intermediate (or secondary) traits and modeled with a multi-trait GP model to predict wheat grain yield [17–18]. In the study of [18], grain yield was predicted for untested environment in which phenotypic data of a target population was not available. The prediction accuracy, however, was not improved with a multi-trait GP model [18]. Compared with multi-trait GP model approach, our two-step approach has a good flexibility to model nonlinear relationship among target and intermediate traits through applying a nonlinear model at the second step (e.g. CGM as in IntCGM or Random Forest as in IntML2).

Another benefit of IntCGM was that the range of predicted among-lines variation [i.e., the regression coefficient (slope) of observed versus predicted values of IntCGM] was closer to 1 compared with that of GP (Fig 6). This would be important in breeding programs [30, 62], although it has not been evaluated in recent studies of the prediction of G×E by GP [8,9,14,16]. In those studies, the accuracy of prediction models was assessed mainly by correlation between predicted and observed (or estimated) values. Although correlation is a good measure of the ordinal accuracy of the prediction (i.e., the accuracy of predicting the order of genotypic values), it does not necessarily reflect the range of genetic variations [63]. In some cases, the accurate prediction of phenotypic values is important for breeding; for example, we may need to maintain the flowering date within a certain range for ease of field management or limit plant height to prevent lodging. When aiming at the application of GP to actual breeding the accurate prediction of the size of genetic variation in a population is as important as the ordinal relationship among genotypes in the population.

## Further improvement of the prediction model

The prediction accuracy of the models was validated using 2-year experiments, which had a 1-month difference in the timing of sowing and planting; one year was used for training, whereas the other year was used as previous researches did [15–16]. Although experiments in 2014 and 2015 were performed in one location, the 2-year experiments were conducted under different environmental conditions (e.g., temperature, day length, and radiation) by employing different cropping seasons. However, other environmental factors, such as soil condition, were fixed in these experiments. To apply our models to a dataset with multiple locations and years, we should take into account other environmental factors, such as soil condition, water supply, and cultivation management, in the models.

In this study the biomass was selected as the target trait for prediction, but the prediction of yield was more challenging. A possible method of implementing accurate prediction of yield is the use of sophisticated CGMs. The potential of several CGMs, such as APSIM [64], has been already demonstrated in practical applications. However, certain complexities may create problems. One of the problems is the accumulation of errors: the errors of parameter estimation would be large if the model includes several parameters. Therefore, models must be simplified in ways such as the use of machine learning (IntML) or variable selection. A sensitivity analysis will be effective to select modules of the models in which variables with little influence on target traits will be distinguished.

Another problem is the increased effort required for measuring plant growth if a model requires a large number of growth parameters. Parameter estimation is one effective solution [15,19,27]. Through these methods, we may be able to omit the measurement of some growth-related traits and to estimate them as parameters in a CGM while measuring the remaining traits in the field. The use of high-throughput phenotyping is another way to enable plant growth to be measured in detail. For example, LAI [65–66] and biomass [67–68] can be measured in a non-destructive way by remote sensing with unmanned aerial vehicles. Such techniques would enable us to measure various kinds of growth-related traits continuously during growth. GP and high-throughput phenotyping technologies could revolutionize breeding [69].

Moreover, the use of a deterministic model in IntCGM may reduce phenotyping costs for the target traits. In IntCGM, the phenotypic values of biomass in the training data were used only for scaling the model's prediction values onto the phenotypic values with $\tau$ as the scaling parameter. Using $\tau$, the RMSE of biomass in known environments decreased by 45% and 68% in 2014 and 2015, respectively. However, the scaling procedure (i.e., the training of model with the phenotypic values of biomass) was not necessary with the use of the prediction values for selecting superior genotypes because the correlation between the predicted and genotypic values of biomass did not change with scaling. This is because the CGM used in this study was deterministic and did not include any parameters to be estimated other than $\tau$. This is another great advantage of IntCGM because the model does not require the phenotypic data of biomass, which in turn requires the laborious destructive measurements of plants.

### Toward application for breeding

In this study, we validated our method with the dataset of the 2-year experiments, which had a 1-month difference in their timings of sowing and planting to simulate different environmental conditions. Although the validation is insufficient to evaluate the potential of the method, our models may be applicable to multi-location-multi-year dataset because CGM is expected to describe G×E when it has an appropriate model structure and the necessary environmental factors. Thus, IntCGM may enable accurate prediction of phenotypes in each target environment and accelerate the development of varieties having excellent viability in the target environments.

Our models may also help to explain the mechanisms causing G×E effects on yield-related traits because they can predict the effects physiologically through CGMs. The predicted values of growth-related "intermediate traits", as well as of yield-related traits, allow us to understand how environmental factors affect growth and have a large impact on yield. This understanding will be of benefit to the mechanical evaluation of environmental characteristics of locations and the appropriate choice of locations used in METs.

### Supporting information

**S1 Fig. Genetic map of SNP markers of RILs.**
(TIF)

**S2 Fig. Comparison of prediction accuracy of panicle weight.** The result of prediction of tested (left) and untested (right) environments are shown. LASSO was chosen as a representative GP model. Three indices were used: Correlation coefficient (r), RMSE (root mean squared error), and slope of the regression line for predicted and observed values. Error bars represent ± 1 s.d. Letters above the bars indicate significant differences determined using the Steel–Dwass test ($p < 0.01$).
(TIFF)

**S1 Table. List of information of genetic markers.**
(CSV)

**S2 Table. List of names of 112 Japanese cultivars used to estimate growth parameters of heading date.**
(TXT)

**S3 Table. Results of ANOVA test of observed traits to detect the effect of G×E.**
(CSV)

**S4 Table. Calculation of wide-sense heritability of observed traits using replicates.**
(CSV)

## Acknowledgments

We thank Hironori Wakabayashi and Koji Watanabe for managing the field experiments. We also thank Takashi Harigae, Teruyo Omura, Miyuki Ishibashi, Noriko Kimoto and Chie Muto for assisting the field measurements. The authors thank Maya Watanabe for organizing the MET data and for conducting the initial analysis of the CGM.

## Author Contributions

**Conceptualization:** Yusuke Toda, Hiroyoshi Iwata.

**Data curation:** Hitomi Wakatsuki, Hiromi Kajiya-Kanegae, Kaworu Ebana.

**Formal analysis:** Yusuke Toda, Toru Aoike.

**Funding acquisition:** Masanori Yamasaki, Kaworu Ebana, Takeshi Hayashi, Hiroshi Nakagawa, Toshihiro Hasegawa, Hiroyoshi Iwata.

**Investigation:** Hiromi Kajiya-Kanegae, Masanori Yamasaki, Takuma Yoshioka, Kaworu Ebana, Takeshi Hayashi, Hiroshi Nakagawa, Toshihiro Hasegawa.

**Methodology:** Yusuke Toda, Hiroyoshi Iwata.

**Project administration:** Hiroyoshi Iwata.

**Resources:** Hitomi Wakatsuki, Masanori Yamasaki.

**Software:** Yusuke Toda, Toru Aoike.

**Supervision:** Hiroyoshi Iwata.

**Validation:** Yusuke Toda.

**Visualization:** Yusuke Toda.

**Writing – original draft:** Yusuke Toda.

**Writing – review & editing:** Yusuke Toda, Hitomi Wakatsuki, Toru Aoike, Hiromi Kajiya-Kanegae, Masanori Yamasaki, Kaworu Ebana, Takeshi Hayashi, Hiroshi Nakagawa, Toshihiro Hasegawa, Hiroyoshi Iwata.

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
