## [Decision Letter · Decision Letter 0]

11 Dec 2019

PONE-D-19-23130

Predicting G×E in biomass of rice: modeling method combining crop growth models and genomic prediction models

PLOS ONE

Dear Dr. Iwata,

Thank you for submitting your revised manuscript to PLOS ONE. I apologize for the delay in returning this manuscript to you. I was able to secure only one external review, and I reviewed the article myself. We invite you to submit a revised version of the manuscript that addresses the points raised during the review process.

Please address both the reviewers and my concerns.

My major concern was with the clarity of the work. The major contribution is the combination of genomic selection and a crop growth model to predict biomass. The approach is quite complex (Figure 2), but the logic is straightforward. Biomass is very strongly correlated with leaf area accumulated prior to flowering time. Both growth rate and flowering time have genetic components. Thus, genomic prediction can predict differences in biomass through these "intermediate traits." The key point seems to be that direct genomic prediction of biomass  is worse (though not that much worse) than the integration the crop growth models. The discussion would benefit from expanding on why using the growth models usually leads to better predictions. I would also include the analysis of yield mentioned in the discussion in results.

I was surprised that r of the predicted values vs untested environment values were so high (Figure 6). It seems the model trains on the means/ lsmeans from two very different replicates in each year. Perhaps this approach is leading to high correlations. How does the model perform if predictions are tested across the single replicates in the following year? This work would be more likely used in this context it seems.

The rationale for the other work reported in the manuscript was unclear to me.  The discussion first describes how rice growth differed between 2014 and 2015. I'd omit this. The discussion then discusses Figure 6 which has the results of the growth model/ genomic prediction. If the other reported data made a scientific contribution, I would discuss them more fully or remove them.

We would appreciate receiving your revised manuscript by Jan 25 2020 11:59PM. To enhance the reproducibility of your results, we recommend that if applicable you deposit your laboratory protocols in protocols.io, where a protocol can be assigned its own identifier (DOI) such that it can be cited independently in the future. For instructions see: http://journals.plos.org/plosone/s/submission-guidelines#loc-laboratory-protocols

We look forward to receiving your revised manuscript.

Kind regards,

Lewis Lukens

Academic Editor

PLOS ONE

Journal Requirements:

**When submitting your revision, we need you to address these additional requirements:**

**Please ensure that your manuscript meets PLOS ONE's style requirements, including those for file naming. The PLOS ONE style templates can be found at http://www.plosone.org/attachments/PLOSOne_formatting_sample_main_body.pdf and http://www.plosone.org/attachments/PLOSOne_formatting_sample_title_authors_affiliations.pdf**Please ensure that you refer to Figure 5 in your text as, if accepted, production will need this reference to link the reader to the figure.

Reviewers' comments:

Reviewer's Responses to Questions

**Comments to the Author**

1. Is the manuscript technically sound, and do the data support the conclusions?

Reviewer #1: Yes

2. Has the statistical analysis been performed appropriately and rigorously? 

Reviewer #1: Yes

3. Have the authors made all data underlying the findings in their manuscript fully available?

Reviewer #1: Yes

4. Is the manuscript presented in an intelligible fashion and written in standard English?

Reviewer #1: Yes

5. Review Comments to the Author

Reviewer #1: 2019-11-19

I have read the revised manuscript and authors’ responses to reviewer’s comments, and thank that the manuscript is well prepared and the responses are almost adequate. I have two questions and would like the authors to address in their Discussion section.

The first is related to the reviewer 4’s question on how to deal with non-repeatable genotype by environment interaction (GE). This is a very good question for any study to be useful for breeding. The authors response was to use “historical environmental data or simulated (assumed) future environmental data.” I think this is a very important point and needs to be emphasized if their models are to be used in assisting breeding. Indeed, GE has always been a challenge to plant breeding, and it is important to differentiate repeatable vs. non-repeatable GE and deal with them differently. In plant breeding, no-repeatable GE is dealt with by testing at multiple locations for multiple years so as to represent the population of environments in the target environment (Yan, 2016 Crop Science). For reliable genomic prediction (GP), genomic models must also be developed based on phenotypic data from such multi-location multi-year trials (Yan et al., 2019, Crop Breeding, genetics, and Genomics). This also applies to prediction models that integrate GP and physiological models. Furthermore, models integrating multi-location multi-year data have to be validated by similarly obtained phenotypic data. It remains a question whether an “integrated” model actually predicted better than GP alone.

My second question/comment is how far does the crop simulation models have to go? In the current study, leaf number, tiller number, and leaf size were simulated using daily temperature and day length to simulate biomass. However, the complexity of the physiological and biological process leading to biomass or grain yield is unlimited. For example, should we go by hourly temperature? Should we also consider CO2, water, and mineral nutrient availability in the models? Is there an optimum level of complexity to target in integrated models?

6. PLOS authors have the option to publish the peer review history of their article (what does this mean?). If published, this will include your full peer review and any attached files.

Reviewer #1: Yes: Weikai Yan

---

## [Author Response · Author response to Decision Letter 0]

2 Feb 2020

> My major concern was with the clarity of the work. The major contribution is the combination of genomic selection and a crop growth model to predict biomass. The approach is quite complex (Figure 2), but the logic is straightforward. Biomass is very strongly correlated with leaf area accumulated prior to flowering time. Both growth rate and flowering time have genetic components. Thus, genomic prediction can predict differences in biomass through these "intermediate traits." The key point seems to be that direct genomic prediction of biomass is worse (though not that much worse) than the integration the crop growth models. The discussion would benefit from expanding on why using the growth models usually leads to better predictions.

The improvement in the prediction accuracy of our model was partly caused by the prediction of heading date, which was included in both IntCGM and IntML. Because we considered the interaction between genotypes on one hand and environmental factors the other, the predicted values of heading date as intermediate traits can contribute to explain the pattern of G×E in biomass. We added the related descriptions to explain the reason why the new models (IntCGM and IntML) performed better than GP (L388-393). Another benefit to use IntCGM has been described in a later part (L425-450).

> I would also include the analysis of yield mentioned in the discussion in results.

As you suggested, we moved the description about the prediction of panicle weight (L470-L474 in old manuscript) to the result part (L370-373). Also, we provide S2 Figure to describe the prediction accuracy of panicle weight (The figure caption was added in L733-738).

> I was surprised that r of the predicted values vs untested environment values were so high (Figure 6). It seems the model trains on the means/ lsmeans from two very different replicates in each year. Perhaps this approach is leading to high correlations. How does the model perform if predictions are tested across the single replicates in the following year? This work would be more likely used in this context it seems.

We added the table of narrow-sense heritability calculated by replicates (S4 Table). We, however, did not test performances of the models when they are trained with a single replicate, but it is clear that the prediction accuracy of all the models decreases at a certain level. Because the replicates were cultivated in the same environment (at least same in environmental variables used in this study), the differences in phenotypic values between the replicates cannot be explained with either genotype data or environmental data. Thus, the separate modelling with a single replicate will only enlarge noises to use a fewer number of replicates for training models, and thus leads to decreases in the prediction accuracy for all the models.

> The rationale for the other work reported in the manuscript was unclear to me. The discussion first describes how rice growth differed between 2014 and 2015. I'd omit this.

As you suggested, we removed the paragraph “Comparison of crop traits between different temperature patterns” (L378-385 in original manuscript).

> The discussion then discusses Figure 6 which has the results of the growth model/genomic prediction. If the other reported data made a scientific contribution, I would discuss them more fully or remove them.

We described about three indices (r, RMSE and slope) in the paragraph (L388-400 in original manuscript). We removed the notation about the slope because it was duplicated with a later paragraph (L436-450). Also, we removed the notation about RMSE (L396-398 in original manuscript). The RMSE of IntCGM in the prediction under the untested environment showed different tendencies in the prediction in the untested environment (Fig 6), but the reason was due to the specific structure of crop growth model we used. Because the explanation for the reason will be so long but have a little generality for other studies, we decided to remove the description. Finally, the description about r was shortened and moved to L384-388, before a newly added part describing the reason why our integrated models could improve prediction accuracies (L388-393). We believe that this revision made the aim of this paragraph clearer.

> Reviewer #1: 2019-11-19

> I have read the revised manuscript and authors’ responses to reviewer’s comments, and thank that the manuscript is well prepared and the responses are almost adequate. I have two questions and would like the authors to address in their Discussion section.

> The first is related to the reviewer 4’s question on how to deal with non-repeatable genotype by environment interaction (GE). This is a very good question for any study to be useful for breeding. The authors response was to use “historical environmental data or simulated (assumed) future environmental data.” I think this is a very important point and needs to be emphasized if their models are to be used in assisting breeding.

> Indeed, GE has always been a challenge to plant breeding, and it is important to differentiate repeatable vs. non-repeatable GE and deal with them differently. In plant breeding, non-repeatable GE is dealt with by testing at multiple locations for multiple years so as to represent the population of environments in the target environment (Yan, 2016 Crop Science). For reliable genomic prediction (GP), genomic models must also be developed based on phenotypic data from such multi-location multi-year trials (Yan et al., 2019, Crop Breeding, genetics, and Genomics). This also applies to prediction models that integrate GP and physiological models.

As you suggested, we viewed our result from the aspect of the decomposition of G×E. We assume that non-repeatable G×E can decompose G×E in variations explained by environmental factors and ones unexplained by environmental factors; the former and latter are referred to as explainable and non-explainable G×E, respectively. The explainable G×E was because of differences in environments between years, whereas the latter was not explained by any obvious environmental factors, therefore, treated as noise. Improvement of prediction accuracies of models in the tested environment indicate that the integrated models can predict the explainable G×E variations. Furthermore, the improvement of accuracies in the untested environment indicate that the integrated models can predict the repeatable G×E variations, because difference between the field trials in two years, where the patterns of temperature and day length differed, was as large as those between locations.

We added a discussion about this topic (L486-503) with a reference you noted [64] (L728-729).

> Furthermore, models integrating multi-location multi-year data have to be validated by similarly obtained phenotypic data. It remains a question whether an “integrated” model actually predicted better than GP alone.

Indeed, the performances of our integrated models were not tested with multi-location data. However, as emphasized in the method part (L103-109), sowing and transplanting were performed in different months between years to produce results under different conditions of day length and temperature. Because of different cultivation periods during 2014 and 2015, the 2-year experiments were not simply yearly replications but were expected to induce different growth patterns under different environmental conditions. Thus, as shown in the better prediction accuracies in the untested environment than GP (Fig 6), it is certain that our models predicted better in the prediction under different patterns of day length and temperature. 

> My second question/comment is how far does the crop simulation models have to go? In the current study, leaf number, tiller number, and leaf size were simulated using daily temperature and day length to simulate biomass. However, the complexity of the physiological and biological process leading to biomass or grain yield is unlimited. For example, should we go by hourly temperature? Should we also consider CO2, water, and mineral nutrient availability in the models? Is there an optimum level of complexity to target in integrated models?

As we wrote in the discussion part (L466-468), it should be avoided to use a crop growth model with too much complexity. However, it is an important problem to find an optimum model with an appropriate number of parameters. We should omit unnecessary variables depending on the target dataset. A sensitivity analysis will be effective to select modules of the models in which variables with little influence on target traits will be distinguished. We added a description of that point (L468-471).

---

## [Editor Report · Decision Letter 1]

20 Feb 2020

PONE-D-19-23130R1

Predicting G×E in biomass of rice: modeling method combining crop growth models and genomic prediction models

PLOS ONE

Dear Dr. Iwata,

Thank you for submitting your manuscript to PLOS ONE. After careful consideration, we feel that it has merit but does not fully meet PLOS ONE’s publication criteria as it currently stands. Therefore, we invite you to submit a revised version of the manuscript that addresses the points raised during the review process.

We would appreciate receiving your revised manuscript by Apr 05 2020 11:59PM. To enhance the reproducibility of your results, we recommend that if applicable you deposit your laboratory protocols in protocols.io, where a protocol can be assigned its own identifier (DOI) such that it can be cited independently in the future. For instructions see: http://journals.plos.org/plosone/s/submission-guidelines#loc-laboratory-protocols

We look forward to receiving your revised manuscript.

Kind regards,

Lewis Lukens

Academic Editor

PLOS ONE

Additional Editor Comments (if provided):

Thank you for the revision. The manuscript is interesting and has improved. My major concern remains with the clarity of the work. For example, a reader should be able to move from introduction to results and get a sense of the findings. The questions posed in the introduction should specifically addressed in results. Now, results is mostly composed of technical statements. Re-writing the section to contain paragraphs with topic sentences would help improve clarity. The article will have greater impact if it can be readily understood. Specific lines to improve are listed below but please edit broadly.

A central point of the article is that the inclusion of growth-related traits into models enabled G x E effects to be predicted. G X E indicates variation in genotypic responses to environment variation. In this study, it seems genotypes were analyzed in single environments, so there is no variation due to the environment except random error. Within this single environment, some genotypes grow faster and others slower, so genotypes will have different growth rates relative to environmental factors such as temperature, but I would refer to these genotypic differences as genetic effects not G x E. G x E effects could explain differences in genotypes between the two years of the study, but this aspect does not seem to be the focus. Specific text includes lines 38-41; 388-390; 391-393, and elsewhere such as 483. Please clarify.

In previous comments, I wrote, “I was surprised that r of the predicted values vs untested environment values were so high (Figure 6).” My understanding is that genomic prediction accuracies estimated for 2014 lines using 2014 data (“tested”) were less accurate than genomic prediction models trained with 2014 data and used to predict 2015 (“untested”) (Figure 6). This odd result, if correct, still does not seem to be explained.

Specific points to potentially clarify:

Line 32: “environmental data was predicted.” Is this in results?

Line 248. Fourteen traits are mentioned as subjected to genomic prediction, but I did not see a clear summary of these traits.

Line 322 “Tiller length” but number of tillers in the figure. Please define axes in Fig 3.

Line 337 Define what is plotted in this figure.. e.g. adjusted line means?

Line 402 “An advantage” over what other option? Why specifically is treating parameters as representations of actual crop conditions beneficial? 

Line 434. The text notes the authors’ approach could save time because biomass is destructive and hard to obtain. This point seems overstated. The intermediate traits themselves do not seem easy to measure. Please clarify. In addition, is biomass a useful trait for rice breeders/ others? If so, please state. This point about the usefulness of the approach was also brought up by the reviewer in terms of validating the model. How much confidence can one have from an analysis of two locations of data?

Line 436 “The prediction accuracy of our models was better…” At times, they are equal (Fig 6).

---

## [Author Response · Author response to Decision Letter 1]

19 Apr 2020

> My major concern remains with the clarity of the work. For example, a reader should be able to move from introduction to results and get a sense of the findings. The questions posed in the

introduction should specifically addressed in results. Now, results is mostly composed of technical statements. Re-writing the section to contain paragraphs with topic sentences would help improve clarity. The article will have greater impact if it can be readily understood. Specific lines to improve are listed below but please edit broadly.

As you pointed out, we revised our manuscript in the following points.

L71-79: We clarified what was remained to be improved in recent studies of the integrated models of GP and CGM.

L80-90, L97-116: We stated about the originality of our research following to the description of use of growth-related traits (L80-90). This revision enabled readers to easily understand why we chose two-step approach. Also, we modified later paragraphs to take consistency of the manuscript (L97-116).

L196: We moved a description of 10-fold cross-validation (L344 in the original manuscript), which should be placed in the material and methods part.

L349-351, L352-L353, and L374-376: We added explanations about topics of each paragraph for whom move to the result part from the introduction part directly.

L378-379: We added a necessary explanation about the result.

L381: We changed the expression for easy understandings.

L400-404: We re-written the paragraph to make the result clearer.

L470-471: We added a description about recent studies of multi-trait GP to clarify the improvement of our study.

> A central point of the article is that the inclusion of growth-related traits into models enabled G x E effects to be predicted. G X E indicates variation in genotypic responses to environment variation. In this study, it seems genotypes were analyzed in single environments, so there is no variation due to the environment except random error. Within this single environment, some genotypes grow faster and others slower, so genotypes will have different growth rates relative to environmental factors such as temperature, but I would refer to these genotypic differences as genetic effects not G x E. G x E effects could explain differences in genotypes between the two years of the study, but this aspect does not seem to be the focus. Specific text includes lines 38-41; 388-390; 391-393, and elsewhere such as 483. Please clarify.

As you pointed out, we edited our manuscript in the following points.

L26-30, L38-39: We changed expressions in the abstract to take consistency with the following revisions.

L55-56, L59-60, L62-63, and L66-67: We changed expressions of explanations about recent researches in the introduction part to make it correspond with revisions in the discussion part.

L420-426: We discussed the improvement of the prediction accuracy in IntCGM and IntML from an aspect of the use of the intermediate traits, not from an aspect of prediction of G×E.

L536-543: We placed a paragraph to discuss the future possibility of prediction of G×E with IntCGM, instead of the paragraph in which we argued that IntCGM succeeded in the prediction of G×E.

Also, we offer to change the title to “Predicting biomass of rice with intermediate traits: modeling method combining crop growth models and genomic prediction models” and running title to “Predicting biomass of rice with intermediate traits” (L1-4). We removed the expression of prediction of G×E from them.

> In previous comments, I wrote, “I was surprised that r of the predicted values vs untested environment values were so high (Figure 6).” My understanding is that genomic prediction accuracies estimated for 2014 lines using 2014 data (“tested”) were less accurate than genomic prediction models trained with 2014 data and used to predict 2015 (“untested”) (Figure 6). This odd result, if correct, still does not seem to be explained.

The result is correct. We added a description about this result (L427-442) and two references (L751-756). Also, we modified figure labels of Fig 6 to make it easier to understand which data was used as training or test data. Corresponded revisions in manuscript are in L395, L401, L410.

This intuitively unexpected result might be owing to two reasons. One is the low heritability of biomass in 2014, which led to lower prediction accuracy in the models [58-59]. To reduce the influence of the heritability level on the index of the prediction accuracy (i.e., a correlation coefficient between observed and predicted phenotypes), the value of r was adjusted by dividing it by the square root of genomic heritability. The adjusted values of r became 0.652 and 0.746 for the biomass in 2014 and 2015, respectively, and had smaller differences than the previous r. 

Another reason for the higher biomass prediction accuracy in 2015 is the GS model built with LASSO. In Fig 5, the biomass prediction accuracy was lower in LASSO than in other models in 2014, whereas the result was the opposite in 2015. Polygenic marker effects seemed more dominant in biomass in 2014 than in 2015 because LASSO is not good at capturing the small effects of a large number of variables. In contrast, the estimation of genomic heritability effectively reflects polygene effects. The differences in the characteristics of each estimation method subsequently caused the difference in the adjusted values of r for the biomass in 2014 and 2015.

> Specific points to potentially clarify:

> Line 32: “environmental data was predicted.” Is this in results?

No, environmental data was not predicted in this study. Because the sentence was misleading, we changed the sentence to “Second, the biomass was predicted from these GP-predicted values and the environmental data using machine learning or crop growth modeling.” (L31-32)

> Line 248. Fourteen traits are mentioned as subjected to genomic prediction, but I did not see a clear summary of these traits.

Fourteen traits consisted of six intermediate traits and the biomass for two years. The result of the prediction of these traits are shown in fig 5. We added an explanation about the traits (L275).

> Line 322 “Tiller length” but number of tillers in the figure. Please define axes in Fig 3.

As you mentioned, the trait should be the number of tillers. We corrected the trait name (L349) and apology for the mistake. 

> Line 337 Define what is plotted in this figure.. e.g. adjusted line means?

As you mentioned, we clearly described that the plotted values are “The adjusted mean values of each line” (L361-362).

> Line 402 “An advantage” over what other option? 

It is an advantage over conventional researches of integrated models of GP and CGM. We made the description clearer (L451-452).

> Why specifically is treating parameters as representations of actual crop conditions beneficial?

One benefit is an easiness to understand the influence of the parameters on the target traits because phenotypic data of intermediate traits are available, as written in the original manuscript (L410-412). Another benefit is the improvement in prediction accuracy in a real datasets because the prediction of complex traits such as yield with an integrated model of GP and CGM was so difficult that no improvement in prediction accuracy had been found in the real datasets [16]. We added a description in the introduction part (L71-77) and the discussion part (L458-463).

> Line 434. The text notes the authors’ approach could save time because biomass is destructive and hard to obtain. This point seems overstated. The intermediate traits themselves do not seem easy to measure. Please clarify. 

As you mentioned, the time-series observation of the intermediate traits is also laborious. We moved the paragraph about the scaling parameter τ (L425-435 in the original manuscript) after a paragraph about the measurement of the intermediate traits using high-throughput phenotyping (L522-533). Following the paragraph describing the reduction of the phenotyping cost of the intermediate traits, it is natural to discuss the possibility of reducing the phenotyping cost of destructive sampling.

> In addition, is biomass a useful trait for rice breeders/ others? If so, please state. 

Rice biomass is an important trait for rice breeding, not only by a need of biomass itself to be used as biofuel [31-32], but also as an essential factor to determine grain yield together with harvest index [33-34]. We added the description in the introduction part (L91-92, L95-96) and four references (L668-679).

> This point about the usefulness of the approach was also brought up by the reviewer in terms of validating the model. How much confidence can one have from an analysis of two locations of data?

It is true that the result will be more confident if we test the model in a larger number of environments. However, we think two environments are not insufficient to prove the ability of the models, as previous researches about an integrated model of GP and CGM also tested in two environments [15-16]. We added a description of this point (L492-494).

> Line 436 “The prediction accuracy of our models was better…” At times, they are equal (Fig 6).

Your indication is correct. Because this notation is repeated in the result and discussion parts, we erased the sentence.

---

## [Editor Report · Decision Letter 2]

18 May 2020

Predicting biomass of rice with intermediate traits: modeling method combining crop growth models and genomic prediction models

PONE-D-19-23130R2

Dear Dr. Iwata,

We are pleased to inform you that your manuscript has been judged scientifically suitable for publication and will be formally accepted for publication once it complies with all outstanding technical requirements.

With kind regards,

Lewis Lukens

Academic Editor

PLOS ONE

Additional Editor Comments (optional):

I apologize for my slow response. I reviewed the article. Thank you for addressing previous concerns.
---

## [Editor Report · Acceptance letter]

26 May 2020

PONE-D-19-23130R2 

Predicting biomass of rice with intermediate traits: modeling method combining crop growth models and genomic prediction models 

Dear Dr. Iwata:

I am pleased to inform you that your manuscript has been deemed suitable for publication in PLOS ONE. Congratulations! Your manuscript is now with our production department. 

With kind regards,

on behalf of

Dr. Lewis Lukens 

Academic Editor

PLOS ONE